# Role of Beta Cell Function and Insulin Resistance in the Development of Gestational Diabetes Mellitus

**DOI:** 10.3390/nu14122444

**Published:** 2022-06-13

**Authors:** Jonas Ellerbrock, Benthe Spaanderman, Joris van Drongelen, Eva Mulder, Veronica Lopes van Balen, Veronique Schiffer, Laura Jorissen, Robert-Jan Alers, Jeanine Leenen, Chahinda Ghossein-Doha, Marc Spaanderman

**Affiliations:** 1School for Oncology and Developmental Biology (GROW), Maastricht University, 6229 ER Maastricht, The Netherlands; mulder_evelien@hotmail.com (E.M.); veronique.schiffer@maastrichtuniversity.nl (V.S.); laura.jorissen@mumc.nl (L.J.); robertjan.alers@mumc.nl (R.-J.A.); c.ghossein@zuyderland.nl (C.G.-D.); marc.spaanderman@mumc.nl (M.S.); 2Department of Obstetrics and Gynecology, Zuyderland Medical Center, 6419 PC Heerlen, The Netherlands; 3Department of Obstetrics and Gynecology, Radboud University Medical Center, 6500 HB Nijmegen, The Netherlands; benthespaanderman99@gmail.com (B.S.); joris.vandrongelen@radboudumc.nl (J.v.D.); 4Department of Obstetrics and Gynecology, Maastricht University Medical Center, 6229 ER Maastricht, The Netherlands; veronica.lopesvanbalen@mumc.nl; 5Department of Finance, Zuyderland Medical Center, 6419 PC Heerlen, The Netherlands; j.leenen@zuyderland.nl; 6Department of Cardiology, Maastricht University Medical Center, 6229 HX Maastricht, The Netherlands

**Keywords:** HOMA-IR, HOMA-β, GDM, OGTT, gestational diabetes mellitus, insulin resistance, beta cell function

## Abstract

Background: Gestational diabetes mellitus (GDM) is a pregnancy complication characterized by second trimester hyperglycemia. Untreated, GDM is related to an increased risk for adverse pregnancy outcomes. Both beta cell dysfunction and insulin resistance underlie impaired glucose tolerance. Understanding the dominant mechanism predisposing to GDM may be important to provide effective treatment in order to improve perinatal outcomes. We hypothesize that insulin resistance rather that beta cell dysfunction predisposes to GDM. Methods: A 75g oral glucose tolerance test (OGTT) was performed on 2112 second-trimester pregnant women to determine the relationship between insulin resistance (HOMA-IR), beta cell function (HOMA-β), and the prevalence of abnormal glucose handling. Results: High insulin resistance raised the risk of GDM (relative risk (RR) 6.1, 95% confidence interval (CI) (4.4–8.5)), as did beta cell dysfunction (RR 3.8, 95% CI (2.7–5.4)). High insulin resistance, but not beta cell function, enhances the necessity for additional glucose lowering medication on top of a low carbohydrate diet in women diagnosed with GDM. Conclusions: Both high insulin resistance and beta cell dysfunction increase the risk of GDM. As increased insulin resistance, rather than beta cell function, is related to an insufficient response to a low carbohydrate diet, we speculate that insulin sensitizers rather than insulin therapy may be the most targeted therapeutic modality in diet-insensitive GDM.

## 1. Introduction

Gestational diabetes mellitus (GDM), defined by impaired glucose handling, first occurring or recognized during gestation, is one of the most prevalent gestational complications affecting 1 to 14% of all pregnancies [1].

During normal pregnancy, adjustments in maternal glucose handling ensure adequate glucose transport towards the fetus. As gestation progresses, sensitivity to insulin attenuates, causing a state of insulin resistance. This state of insulin resistance is primarily initiated by a rise in maternal and placental anti-insulinergic hormones, including progesterone, cortisol, tumor necrosis factor α (TNF-α), placental growth hormone, and a fall in plasma adiponectin. Concomitantly, the rise in human placental lactogen (hPL) and prolactin enhances maternal beta cell mass and glucose-stimulated insulin secretion, maintaining steady glucose control homeostasis, despite the loss in insulin sensitivity [2].

GDM is thought to originate when beta cell function cannot be sufficiently upregulated relative to the increased insulin resistance [3]. On the contrary, an excessive gestational rise in insulin resistance with normal beta cell function also relates to disturbed glucose handling [4]. As maternal glucose transport to the fetus is concentration-dependent and insulin-independent, increased plasma levels raise fetal glucose availability, leading to increased fetal growth and metabolic needs [2]. When undetected or untreated, GDM is associated with an increased risk of maternal gestational hypertensive disorders, caesarean section, and neonatal complications including fetal macrosomia, shoulder dystocia, and neonatal hypoglycemia [5,6]. Treatment of GDM reduces these complications, providing similar neonatal outcomes as nondiabetic pregnancies [5]. Timely detection and effective treatment are therefore crucial. 

Widely used treatment options for GDM include dietary intervention, insulin supplementation, and metformin therapy. Each of these therapies target one or more of the interacting factors affecting glucose homeostasis. Dietary carbohydrate restriction is instituted to a lower exogenous influx of glucose, while insulin therapy compensates for beta cell dysfunction and insulin resistance, and endogenous glucose synthesis can be improved by metformin [7]. These different underlying mechanisms raise the question of how insulin resistance and beta cell function relate to each other in women with GDM and whether therapy could be targeted to the underlying disturbing mechanism. 

In this study, we tested the hypothesis that insulin resistance rather than beta cell dysfunction underlies GDM. 

## 2. Materials and Methods

### 2.1. Study Population

This cohort study was carried out at the department of Obstetrics and Gynecology of Maastricht University Medical Center, Maastricht, The Netherlands, between January 2014 and December 2019. 

Pregnant women were recruited when undergoing an oral glucose tolerance test (OGTT) between 24–28 weeks of gestation. According to our national guidelines, an OGTT is offered to women at assumed increased risk for glucose handling disorders. Indications are body mass index (BMI) > 25 kg/m^2^, a family history of diabetes, polycystic ovary syndrome (PCOS), a history of macrosomia (a previous neonate >90th percentile or >4.250 g), or GDM and non-Northern European ancestry. 

Pregnant women were excluded when meeting the following criteria: diagnosis of diabetes and/or hypertension before the current pregnancy, incomplete diagnostic information because of vomiting during the test or missing medical records, gestational age <22 or >30 weeks, or undergoing OGTT for diagnostic reasons (i.e., macrosomia, hydramnios, or abnormal random glucose). 

### 2.2. Anthropometric Data 

Ethnicity and pre-pregnancy weight were self-reported. Height was measured at the time of the OGTT. Height and pre-pregnancy weight were used to calculate pre-pregnancy body mass index (pre-BMI). Blood pressure was measured in a quiet environment in a sitting position, using an osscilometric method (Carescape V100, GE Healthcare, Waukesha, WI, USA) for 30 min at a 3 min interval. The median value was reported and was used for the analysis.

### 2.3. Biochemical Analysis 

Maternal venous blood samples were drawn in the morning after an overnight fast of ≥9 h in order to assess the metabolic profile. The following (metabolic) parameters were assayed: fasting plasma glucose (FPG), fasting plasma insulin (FPI), triglycerides (TG), low-density lipoprotein (LDL), high-density lipoprotein (HDL), and glycated hemoglobin (HbA1c). Venous blood sampling was followed by the ingestion of a 75 g glucose drink (corresponding to 82.5 g Dextrose monohydrate from Fagron, Rotterdam, The Netherlands). Additional blood samples were collected for plasma glucose 1 and 2 h post load. Lipid samples, urine protein, and plasma glucose were measured using an autoanalyser (Cobas 8000 Roche, Basel, Switzerland). Fasting serum insulin levels were measured using an immune-assay (Immulite XPi, Siemens Healthineers, Erlangen, Germany). 

Insulin resistance and beta cell function were determined by calculating the Homeostatic Model Assessment for insulin resistance (HOMA-IR) and the beta cell function (HOMA-β) index with the following formulas [8,9];
(1)$$\mathrm{HOMA}-\mathrm{IR}=\frac{\left[\mathrm{fasting}\;\mathrm{insulin}\;\left(\mathrm{pmol}/L\right)\right]\times \left[\mathrm{fasting}\;\mathrm{glucose}\;\left(\mathrm{mmol}/L\right)\right]}{135}\;$$
(2)$$\mathrm{HOMA}-\beta =\frac{20\times \left[\mathrm{fasting}\;\mathrm{insulin}\;\left(\mathrm{pmol}/L\right)\right]}{[\mathrm{basal}\;\mathrm{glucose}\;\left(\mathrm{mmol}/L\right)-3.5}\;\%$$


We considered women insulin resistant or having a low beta cell function when their HOMA-IR exceeded 1.5 multiples the mean (MoM) or their HOMA-β was below 0.5 MoM, respectively, of that of healthy non-gestational diabetic second trimester women [10,11]. Low and high insulin resistance, 0.5 and 1.5 MoM-IR, corresponded with 5.3 and 15.9, respectively, while low and high beta cell function, 0.5 and 1.5 MoM-β, corresponded with with 134 and 403, respectively.

### 2.4. Adverse Pregnancy Outcomes 

GDM was diagnosed according to the International Association of Diabetes in Pregnancy Study Group (IADPSG) criteria [12]. In these criteria, GDM was diagnosed when the fasting glucose was >5.1 mmol/L, 1-h plasma glucose > 10.0 mmol/L, and 2-h plasma glucose > 8.5 mmol/L. In case GDM was diagnosed, the first dietary treatment was given, lowering the carbohydrate intake. If subsequently, after one week of dieting, the fasting and 1 h post prandial glycemic levels were below 5.3 mmol/L and 7.8 mmol/L, respectively, then the glucose levels were tested monthly. If at each assessment point the women did not meet the established levels of glycemic control, then metformin therapy was started. In the weeks following, glucose curves were monitored and dosage was adjusted if sufficient glycemic control was not reached. If both diet and metformin were unable to help achieve sufficient glycemic control, metformin was replaced by insulin or insulin was added. 

### 2.5. Statistical Analysis 

The statistical analysis was conducted using SPSS (Version 24.0, IBM Corp, Armonk, NY, USA). Quantitative values were expressed as mean with standard deviation (SD), and the mean differences were tested with an independent sample *T*-test. ANOVA and Chi-Square tests or Fisher Exact test along with a risk ratio, whenever applicable, were used to evaluate univariate differences between continuous and categorical variables, respectively. A *p*-value of less than 0.05 was considered significant.

In order to further explore the role of insulin resistance and beta cell function related to abnormal glucose handling and the use of blood glucose lowering medication, metformin, insulin, or a combination of both, the groups were categorized in quintiles of increasing insulin resistance and beta cell function. The effect of increasing the beta cell function and insulin resistance in the presence of GDM and the use of blood glucose lowering therapy were tested using logistic regression analysis. 

This manuscript was prepared with due regard to the Strengthening the Reporting of Observational Studies in Epidemiology (STROBE) statement [13].

## 3. Results

A total of 2112 pregnant women were included. Of these, 576 (27.3%) were diagnosed with GDM, whereas the remaining 1536 had normal glucose tolerance. Baseline maternal age, height, pre-pregnancy weight and pre-pregnancy body mass index (pre-BMI), and a history of PCOS were significantly different between the two groups, as shown in Table 1. 

Weight, BMI, mean arterial pressure (MAP), triglyceride, and HbA1c in the second trimester were significantly higher in the GDM group, while total cholesterol and HDL were significantly lower. Compared with the control group, the fasting insulin levels were significantly higher. HOMA-IR was significantly higher in women with GDM, while HOMA-β was significantly lower. 

The relationship between HOMA-IR relative to HOMA-β and the incidence of GDM is shown in Table 2. The prevalence of GDM in women with low beta cell function significantly increased when HOMA-IR increased. Moreover, an increase in beta cell function (HOMA-β) was correlated with significantly less GDM for each HOMA-IR group. From a functional perspective, compared with women with normal insulin resistance and beta cell function, in women with increased insulin resistance and normal beta cell function, abnormal glucose handling occurred six-fold more often (relative risk (RR) 6.1, 95% confidence interval (CI) (4.4–8.5)), while in women with normal insulin resistance and low beta cell function, the risk for abnormal glucose handling was almost four-fold (RR 3.8, 95% CI (2.7–5.4)).

In order to explore the distribution of HOMA-IR and HOMA-β, the study population diagnosed with GDM was categorized into quintiles: Q1Q1 represents the group of women with the lowest HOMA-IR and lowest HOMA-β, while Q5Q5 denotes women with the highest insulin resistance and highest beta cell function. In all beta cell function quintile groups, as insulin resistance increased, the groups consisted of significantly more women with GDM. Moreover, in all insulin resistance quintile groups, an increase in beta cell function was correlated with a significant decrease in GDM (Table 3).

As shown in Table 4, 81% of all women with abnormal glucose tolerance achieved glycemic control through dietary intervention only, while 19% needed glucose lowering medication. 

For the three lowest quintiles of beta cell function, medication use was significantly higher as insulin resistance increased. In contrast, medication use was only significantly lower with increasing beta cell function in the highest quintile of insulin resistance.

## 4. Discussion

In healthy gestation, the increased insulin resistance is counterbalanced by an upregulation of the beta cell function, maintaining normoglycemia. Our population showed, in contrast with insulin resistance, that low beta cell function was not more prevalent among GDM, suggesting no clinically significant contribution of beta cell function in abnormal glucose handling. However, increased beta cell function is related to improved glycemic control and significantly less GDM. As insulin resistance increases, the relative contribution of increased beta cell function in the prevention of GDM decreases. As the largest group of women in the study population have high insulin resistance, the otherwise significant effect of beta cell function regarding the onset of GDM becomes overshadowed. The difference in the contribution of beta cell function to GDM relative to insulin resistance has been described previously and seems to be correlated with obesity [14]. A low beta cell function contribution is more pronounced for hyperglycemia in non-obese women compared with obese women. As most of the women in this study are overweight or obese, the impact of beta cell function on the development of GDM is negligible.

We observed more women have GDM as insulin resistance gradually increases. One in two women diagnosed with GDM have excessive insulin resistance compared with one in five with normal glucose handling. Increased weight and BMI may be a dominant predisposing factor for developing GDM. Obesity is related with decreased insulin-stimulated glucose uptake and metabolism in the skeletal muscle and adipocytes [15,16]. Concurrently, in obese women, downregulation of the glucose transporter type 4 (GLUT4) expression in adipocytes is observed [15]. As such, obesity predisposes defective insulin signaling, lowering insulin sensitivity and trans-membranous glucose transport capacity, ultimately decreasing plasma-glucose removal. 

Furthermore, we found that women with GDM have significant higher triglycerides and lower HDL levels. This is known as diabetic dyslipidaemia and is identified as a consequence of insulin resistance and obesity [17,18,19,20]. Obesity-associated dyslipidaemia is likely related to insulin resistance, which is in agreement with our results.

GDM can mostly be effectively treated with a carbohydrate-restricted diet. Low-carbohydrate diets significantly reduce postprandial glucose levels and improve pregnancy outcomes by reducing the incidence of macrosomia-related caesarean sections and the requirement for insulin therapy [21]. In approximately 20% of women with GDM, dietary intervention is insufficient to reach glycemic control. Notably, the need for adjuvant medication significantly increases when HOMA-IR increases, but does not correlate significantly with HOMA-β. Women who need medication have a significantly higher BMI and insulin resistance is more prevalent. It is therefore plausible that prominent insulin resistance mainly underlies the need for glucose lowering medication, especially when it concerns overweight women. 

Based on the underlying mechanism, women with low beta cell function in combination with low insulin resistance are likely to benefit from insulin therapy. Only a minor group of women with GDM have this physiological profile, as 73% have high insulin resistance. As insulin insensitivity rather than insulin secretion defects underlie GDM, providing insulin as a starting medication would be mechanistically illogical. Instead, metformin will likely be the most effective treatment by improving insulin sensitivity. Studies about the safety and efficacy of metformin in GDM show that metformin is associated with a lower risk of gestational hypertension, pre-eclampsia, macrosomia, and neonatal hypoglycemia compared with insulin therapy [22,23,24], even though the glycemic control profiles are found to be comparable [25,26]. Compared with insulin, metformin is able to attain target glucose levels faster at treatment initiation, lowering fetal exposure to hyperglycemia [27]. On the other hand, long term effects in the offspring of mothers treated with metformin remain unclear, as exposure to metformin is associated with increased offspring weight, but not with height or BMI [28]. Larger follow-up studies are still needed to clarify the long-term metabolic effect of metformin on offspring. 

Apart from this, metformin would be a more targeted treatment compared with insulin. Therefore, it is important to identify the glycemic pathophysiology underlying GDM, so that effective, personalized care may be provided. 

A major strength of this study is the large prospective cohort in which both HOMA-IR and HOMA-β were evaluated. Additionally, this is the first study of this extent to evaluate the relationship between beta cell function and insulin resistance in order to determine the underlying mechanism predisposing GDM. There are several limitations that need to be addressed. First, the study population primary consisted of women of Northern European ancestry, which could affect generalizability. Studies have shown that the prevalence of GDM differs between ethnic groups. Asian women are found to be at higher risk for GDM compared with Caucasian women, despite generally having a lower BMI [29]. Previous literature has shown that the increase in insulin resistance during pregnancy is similar between different ethnic groups, while the increase in beta cell function is significantly lower in Asian women [30]. Asian women have been found to have smaller beta cell mass with reduced insulin secretion capacity [31]. Knowing the ethnic differences in the underlying mechanism leading to attenuated glucose handling, we think that despite the homogenous population, the conceptual finding is still translatable to the general population. Second, we have not evaluated maternal and perinatal outcomes. However, the aim of this study was to investigate underlying mechanisms that predispose GDM, rather than GDM-related pregnancy outcomes. Third, our study may have suffered from selection bias, as we unexpectedly observed higher prevalence of PCOS in the non-GDM group. PCOS is associated with an increased risk of pregnancy complications, including GDM [32]. As PCOS is a heterogenous syndrome, it has a broad spectrum of laboratory manifestations and clinical symptoms [33,34]. As such, a substantial subgroup of women diagnosed with PCOS have insulin resistance comparable to healthy controls [35]. We speculate that in our study population, most women diagnosed with PCOS may belong to this insulin sensitive subgroup. Considering the rather low prevalence of PCOS in our studied population, these findings suggest many women are undiagnosed as having PCOS. Moreover, women with PCOS associated with insulin resistance may have a lower chance of getting pregnant, which may have led to a reduction in the number of women with insulin-resistant PCOS in our study population. 

## 5. Conclusions

High insulin resistance poses the highest risk on impaired glucose handling compared with beta cell dysfunction. Based on the dominant underlying mechanism of GDM, metformin instead of insulin would be the most effective medication option. We speculate that, in addition to an OGTT, fasting insulin could be assessed to estimate beta cell function and insulin resistance in pregnant women so as to enable personalized and effective care. Further clinical research is necessary in order to evaluate this targeted therapy. 

## Figures and Tables

**Table 1 nutrients-14-02444-t001:** Baseline characteristics of the women with a normal or abnormal second-trimester oral glucose tolerance test (OGTT).

	OGTT		
	Normal	Abnormal	*p*
	*n* = 1536	*n* = 576	
Age (y)	31.1 ± 4.7	32.5 ± 5.4	<0.001
Pre-pregnancy Weight (kg)	74.4 ± 16.7	80.7 ± 17.5	<0.001
Pre-pregnancy BMI (kg/m^2^)	26.7 ± 5.7	29.3 ± 5.8	<0.001
Nulliparous (%)	49.7	45.1	0.058
Singleton pregnancy (%)	97.1	97.9	0.283
Northern European Ancestry (%)	79.4	76.3	0.129
History of GDM (%)	2.9	4.3	0.122
Family history of DM (%)	18.8	15.8	0.094
History of macrosomia (%)	7.2	6.8	0.791
PCOS (%)	6.1	2.3	<0.001
Gestational age OGTT (wk^+d^)	25^+2^ ± 1^+0^	25^+2^ ± 1^+1^	0.838
Weight (kg)	80.7 ± 16.2	87.0 ± 16.8	<0.001
BMI (kg/m^2^)	29.0 ± 5.4	31.6 ± 5.8	<0.001
MAP (mmHg)	80 ± 7	83 ± 7	<0.001
Cholesterol (mmol/L)	6.2 ± 1.1	6.0 ± 1.1	0.002
HDL (mmol/L)	2.1 ± 0.5	1.9 ± 0.4	<0.001
LDL (mmol/L)	3.2 ± 1.0	3.2 ± 1.0	0.107
Triglycerides (mmol/L)	1.98 ± 0.73	2.26 ± 0.82	<0.001
HbA1c (mmol/mol)	4.8 ± 0.3	5.0 ± 0.4	<0.001
OGTT			
Fasting glucose (mmol/L)	4.7 ± 0.3	5.3 ± 0.5	<0.001
Glucose load 1 h (mmol/L)	7.0 ± 1.4	9.3 ± 1.8	<0.001
Glucose load 2 h (mmol/L)	5.9 ± 1.1	7.7 ± 1.6	<0.001
Fasting insulin (pmol/L)	52.7 ± 39.9	79.4 ± 51.5	<0.001
HOMA-IR	11.1 ± 8.5	19.2 ± 13.9	<0.001
HOMA-β (%)	921 ± 707	878 ± 503	0.183
Insulin resistance (%)	19.0	53.4	<0.001
Low beta cell function (%)	52.0	51.8	0.927

y = year; BMI = body mass index; GDM = Gestational diabetes mellitus; DM = Diabetes Mellitus; PCOS = polycystic ovary syndrome; wk^+d^ = gestational age in weeks + days; MAP = mean arterial pressure; HDL = high-density lipoprotein; LDL = low-density lipoprotein; HbA1c = glycated hemoglobin; OGTT = oral glucose tolerance test; HOMA-IR = insulin resistance; HOMA-β = beta cell function. Data are presented as mean ± standard deviation (SD) or as numbers (%).

**Table 2 nutrients-14-02444-t002:** Percentage of women with an abnormal glucose tolerance test (as diagnosis of GDM) related to HOMA-IR and HOMA-β.

		Insulin Resistance (HOMA-IR)			
		<0.5 MoM	0.5–1.5 MoM	>1.5 MoM	
**Beta cell function** **(HOMA-β)**	**<0.5 MoM**	36/372	190/642	64/64	290/1078 *
		(9.7%)	(29.6%)	(100%)	(26.9%)
	**0.5–1.5 MoM**	0/17	35/454	226/480	261/951 *
		(0%)	(7.7%)	(47.1%)	(27.4%)
	**>1.5 MoM**	0/0	0/3	9/43	9/46
		(0%)	(0%)	(20.9%)	(19.6%)
		36/389 *	225/1099 *	299/587 *	560/2075
		(9.3%)	(20.5%)	(39.0%)	(27.0%)

0.5–1.5 multiples the mean (MoM) defines normal glucose tolerance pregnancy ranges of insulin resistance (IR) and beta cell function. <0.5 MoM insulin resistance indicates a low IR, while IR >1.5 MoM indicates a high IR. Beta cell function <0.5 defines low beta cell function, while >1.5 MoM defines high beta cell function. Data are presented as numbers (%). * (*p* < 0.001).

**Table 3 nutrients-14-02444-t003:** Percentage of women with an abnormal second trimester oral glucose tolerance test as a function of insulin resistance and concomitant beta cell function.

		Insulin Resistance (HOMA-IR)					
		Q1	Q2	Q3	Q4	Q5	
**Beta cell function** **(HOMA-β)**	**Q1**	35/306	30/83	11/11	10/10	3/3	89/413 *
		(11.4%)	(36.1%)	(100%)	(100%)	(100%)	(21.5%)
	**Q2**	5/79	24/183	48/104	34/34	7/7	118/407 *
		(6.3%)	(13.1%)	(46.2%)	(100%)	(100%)	(29.0%)
	**Q3**	0/14	4/93	19/150	68/106	49/49	140/412 *
		(0%)	(4.3%)	(12.7%)	(64.2%)	(100%)	(34.0%)
	**Q4**	0/13	1/33	6/108	27/152	78/108	112/414 *
		(0%)	(3%)	(5.6%)	(17.8%)	(72.2%)	(27.1%)
	**Q5**	0/3	1/15	1/39	8/111	87/244	97/412 *
		(0%)	(6.7%)	(2.6%)	(7.2%)	(35.7%)	(23.5%)
		40/415 *	60/407 *	85/412 *	147/413 *	224/411 *	556/2058
		(9.6%)	(14.7%)	(20.6%)	(35.6%)	(54.5%)	(27.0%)

Q = quintile. *p* for trend; * (*p* < 0.001).

**Table 4 nutrients-14-02444-t004:** Percentage of women with GDM who received medication as a function of insulin resistance and concomitant beta cell function.

		Insulin Resistance (HOMA-IR)					
		Q1	Q2	Q3	Q4	Q5	
**Beta cell function** **(HOMA-β)**	**Q1**						17/89 (19%) **
		M: 0 (0%)	M: 3 (10%)	M: 4 (36%)	M: 1 (10%)	M: 1 (33%)	M: 9 (10%)
		I: 0 (0%)	I: 1 (3%)	I: 4 (36%)	I: 0 (0%)	I: 1 (33%)	I: 6 (7%)
		MI: 0 (0%)	MI: 0 (0%)	MI: 0 (0%)	MI: 1 (10%)	MI: 1 (33%)	MI: 2 (2%)
	**Q2**						23/118 (19%) **
		M: 0 (0%)	M: 2 (8%)	M: 6 (13%)	M: 3 (9%)	M: 0 (0%)	M: 11 (9%)
		I: 0 (0%)	I: 0 (0%)	I: 2 (4%)	I: 4 (12%)	I: 2 (29%)	I: 8 (7%)
		MI: 0 (0%)	MI: 1 (4%)	MI: 0 (0%)	MI: 2 (6%)	MI: 1 (14%)	MI: 4 (3%)
	**Q3**						28/140 (20%) **
		M: 0 (0%)	M: 0 (0%)	M: 1 (5%)	M: 2 (3%)	M: 10 (20%)	M: 13 (9%)
		I: 0 (0%)	I: 0 (0%)	I: 2 (11%)	I: 6 (9%)	I: 5 (10%)	I: 13 (9%)
		MI: 0 (0%)	MI: 0 (0%)	MI: 0 (0%)	MI: 0 (0%)	MI: 2 (4%)	MI: 2 (1%)
	**Q4**						22/112 (20%)
		M: 0 (0%)	M: 1 (100%)	M: 0 (0%)	M: 4 (15%)	M: 11 (14%)	M: 16 (14%)
		I: 0 (0%)	I: 0 (0%)	I: 1 (17%)	I: 0 (0%)	I: 4 (5%)	I: 5 (4%)
		MI: 0 (0%)	MI: 0 (0%)	MI: 0 (0%)	MI: 0 (0%)	MI: 1 (1%)	MI: 1 (1%)
	**Q5**						16/97 (16%)
		M: 0 (0%)	M: 0 (0%)	M: 0 (0%)	M: 1 (13%)	M: 13 (15%)	M: 14 (14%)
		I: 0 (0%)	I: 0 (0%)	I: 0 (0%)	I: 0 (0%)	I: 2 (2%)	I: 2 (2%)
		MI: 0 (0%)	MI: 0 (0%)	MI: 0 (0%)	MI: 0 (0%)	MI: 0 (0%)	MI: 0 (0%)
		0/40 (0%)	8/60 (13%)	20/85 (24%)	24/147 (16%)	54/224 (24%) *	106/556 (19%)
		M: 0 (0%)	M: 6 (10%)	M: 11 (13%)	M: 11 (7%)	M: 35 (16%)	M: 63 (11%)
		I: 0 (0%)	I: 1 (2%)	I: 9 (11%)	I: 10 (7%)	I: 14 (6%)	I: 34 (6%)
		MI: 0 (0%)	MI: 1 (2%)	MI: 0 (0%)	MI: 3 (2%)	MI: 5 (2%)	MI: 9 (2%)

Q = quintile, M: metformin, I: insulin and MI: metformin and insulin. *p* for trend; * (*p* < 0.05), ** (*p* < 0.01).

## Data Availability

The data presented in this study are available on request from the corresponding author.

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
