# Peer review of "Role of Beta Cell Function and Insulin Resistance in the Development of Gestational Diabetes Mellitus"

_nutrients, 2022, doi:10.3390/nu14122444_

Round 1

Reviewer 1 Report

Thank you for this interesting and well written paper. I have only some minor suggestions for improvement throughout the text:

Line 46: on the contrary instead of contrary

Line 49: raise instead of raises

Line 75: please, better describe "history of microsomia"

Line 93: 82.5-gram of Dextrose monohydrate by Fagron corresponds to 75 grams of glucose?

Line 95: do you have any data on insulin levels during the OGTT?

Line 95: it would be more precise to write urine protein instead of proteinuria

Line 111: when fasting glucose was...

Line 118: unable to help reaching instead of reach

Line 142. triglycerides instead of triglyceride

Line 162: (p < 0.001).

Line 188: of rather than for 

Line 195: contribution is more pronounced instead of contributes more pronounced

Line 198: negligible instead of neglectable 

Line 199: having instead of have

Line 213: reduce... improve...

Line 218: significantly instead of significant

Line 254: of instead of on

Line 260: chance instead of change

Reviewer 2 Report

REVIEW of the study by Ellerbrock J, Spaanderman B, Van Drongelen J, et al. Role of beta cell function and insulin resistance in the development of gestational diabetes mellitus (GDM).

This is large study (n=2112 women, including 576 women with GDM), where Authors analysed relationship between HOMA-IR (as a marker of insulin resistance -IR) and HOMA-beta (as a marker of beta cell function) and development of GDM. The authors conclude that HOMA-IR (rather than HOMA-beta) is the principal factor related to development of GDM. On the other hand, if GDM develops in a women with low beta cell function then increasing insulin resistance is translated into a greater probability of of failure of dietary treatment, and the need for instigation of medical treatment (insulin sensitizers or inulin).

Furthermore, that Authors suggest that insulin-sensitizers, rather than insulin therapy may be the most targeted therapeutic modality in diet-insensitive GDM.

The study is very interesting and raises very pertinent issues, yet several aspects need to be discussed and clarified.

MAIN POINTS:

1. The Authors should be cautious in concluding that insulin sensitizers might be a better option than insulin, as their study did not address this particular issue. It is true that metformin appears to be safe in GDM (e.g. Guo L. et al.J Diabetes Res . 2019;2019:9804708. doi: 10.1155/2019/9804708. eCollection 2019), but slightly better outcomes of metformin-treated patients, e.g. in terms of pre-eclampsia, macrosomnia, etc, might result from the fact that women with more severe forms of GDM are usually treated with insulin (46.3% of women in Metformin in Gestational Diabetes Trial in 2008 required supplemental insulin, i.e. there was a high metfotmin failure rate). Furthermore, there is an unclear issue regarding effects of metformin on fetal metabolic programming. For instance, it is not clear whether heavier weight of children (about 12-18 months old) whose mother received metformin during pregnancy may have any metabolic consequences (Carlsen SM, et al. Pediatrics 2012; 130(5): e1222-26; Ijas H, Vaarasmaki M, et al. BJOG 2015; 122(7): 994-1000). Notably 8-year old children of metformin-treated mother had higher fasting glucose at the age of 8 years (Ro TB, Ludvigsen HV, et al. Scand J Clin Lab Invest 2012; 72 (2); 570-75).

2. Apart from quartile analysis, is there any value of HOMA-IR above which there is a significant rise in the risk of GDM?

3It is a pity that Authors performed full OGTT, but measured only fasting insulin, as there are several IR indices derived from OGTT (reflecting peripheral, rather than hepatic insulin resistance). Moreover correlation between IR derived from fasting and OGTT values is only moderate, including pregnancy period, with high prevalence of early GDM (Lewandowski K et al. Endokrynol Pol . 2022;73(1):1-7. doi: 10.5603/EP.a2021.0095.). Therefore, conclusions might be different, if other IR indices were taken into account, particularly given that early GDM increases the likelihood of insulin therapy as well as the likelihood of metformin failure (Gante I, Melo D, et al. Eur J Endocrinol Oct 25.pii: EJE-17-0486. https://doi.org/10.1530/EJE-17-0486).

4. I am surprised by such low prevalence of history of PCOS, that is even lower in more overweight women with GDM (2.3% versus 6.1%, p<0.001, table 1). Reported prevalence of PCOS is usually around 8-10% (and even higher if Rotterdam Consensus criteria are used - Hum Reprod. 2004 ;19(1):41-7. doi: 10.1093/humrep/deh098). Therefore, either we are dealing with an extremely reproductively healthy Dutch population, or there is a significant under-reporting of the true prevalence of PCOS, particularly in GDM cohort.

Minor points:

The Authors state that OGTT was performed during the second trimester of pregnancy (Subsection 2.1 Study population). Instead, it should be stated what weeks of pregnancy.
